# Sensory experience during early sensitive periods shapes cross-modal temporal biases

Stephanie Badde[1,2]*, Pia Ley[1], Siddhart S Rajendran[1,3], Idris Shareef[1,3], Ramesh Kekunnaya[1,3], Brigitte Röder[1]

[1]Biological Psychology and Neuropsychology, University of Hamburg, Hamburg, Germany; [2]Department of Psychology and Center of Neural Science, New York University, New York, United States; [3]Child Sight Institute, Jasti V Ramanamma Children's Eye Care Center, LV Prasad Eye Institute, Hyderabad, India

**Abstract** Typical human perception features stable biases such as perceiving visual events as later than synchronous auditory events. The origin of such perceptual biases is unknown. To investigate the role of early sensory experience, we tested whether a congenital, transient loss of pattern vision, caused by bilateral dense cataracts, has sustained effects on audio-visual and tactile-visual temporal biases and resolution. Participants judged the temporal order of successively presented, spatially separated events within and across modalities. Individuals with reversed congenital cataracts showed a bias towards perceiving visual stimuli as occurring earlier than auditory (Expt. 1) and tactile (Expt. 2) stimuli. This finding stood in stark contrast to normally sighted controls and sight-recovery individuals who had developed cataracts later in childhood: both groups exhibited the typical bias of perceiving vision as delayed compared to audition. These findings provide strong evidence that cross-modal temporal biases depend on sensory experience during an early sensitive period.

*For correspondence:
stephanie.badde@nyu.edu

Competing interests: The authors declare that no competing interests exist.

## Introduction

In every moment, a multitude of information reaches our brain through the different senses. These sensory inputs need to be separated, ordered in space and time to derive a coherent representation of the environment. Yet, the perception of temporal order is seldom veridical. Reports illustrating the subjectivity of cross-modal temporal perception date back to 18th and 19th century astronomers; small but stable individual biases in the perceived timing of visual and auditory events caused significant differences in the measurements of stellar transit times which in turn resulted in bitter scientific disputes. These early reports inspired the pioneering work of Wilhelm Wundt and colleagues on cross-modal temporal perception (*Sanford, 1888*; *Canales, 2010*) and, hence, are seen as the origin of experimental psychology (*Mollon and Perkins, 1996*). Early psychologists like William James transferred the personal equation, developed by Bessel to solve the astronomers' problem of stable cross-modal temporal biases by adding a fixed amount to an observer's measurements, to cognitive processes. Yet, even though the relative timing of events is crucial for the perception of time and causality (*Hume, 2012*; *Pöppel, 1988*; *Pöppel, 1997*; *Spence and Squire, 2003*; *Aghdaee et al., 2014*), it has yet remained unsolved why humans consistently misjudge temporal order across different sensory modalities.

Determining the spatio-temporal order of events across sensory modalities poses an especially difficult challenge as information arriving through different senses travels at different speeds – in the environment and within the nervous system (*Hirsh and Sherrick, 1961*). Light travels faster than sound, but within the nervous system, auditory information reaches the brain faster than visual

information (**Fain, 2019**). At a distance of approximately 10 m from the observer, the environmental and the physiological differences presumably cancel each other out (**Pöppel, 1988**). However, both the relative time of arrival at the sensors and neural transmission times vary with the physical properties of the stimulus (**King, 2005**). Curiously, the brain seems to be able to adjust the perception of temporal order for variations in signal strength (**Kopinska and Harris, 2004**) and physical distance (**Engel and Dougherty, 1971**; **Alais and Carlile, 2005**), yet, within their peripersonal space humans typically perceive visual events as delayed compared to sounds (**Keetels and Vroomen, 2012**; **Zampini et al., 2003**; **Grabot and van Wassenhove, 2017**).

Cross-modal temporal biases exhibit a paradoxical characteristic, they are remarkably stable across longer periods of time (**Sanford, 1888**; **Grabot and van Wassenhove, 2017**) while being malleable within short time periods. After humans are exposed to a series of asynchronous stimulus pairs with a constant lag between vision and audition (**Fujisaki et al., 2004**; **Vroomen et al., 2004**) (or vision and touch; **Keetels and Vroomen, 2008**), they perceive the temporal order of subsequent visual-auditory events as shifted in the lag-reversed direction. Such temporal recalibration effects typically last for several minutes. Moreover, on a moment-by-moment basis, the perceived relative timing of events in different modalities changes with the focus of attention. Titchener's law of prior entry (**Titchener, 1908**) states that attention towards one sensory input channel is capable of speeding up perceptual processing of that sensory information resulting in a demonstrable shift of temporal order perception towards events in the attended sensory modality (**Spence et al., 2001**; **Zampini et al., 2005**; **Vibell et al., 2007**; **Spence, 2010**). The co-existence of short-term plasticity and long-term stability re-emphasizes the question that has troubled scientists for 250 years: why do cross-modal temporal biases persist despite the brain's capability for recalibration?

A potential reason for the high stability of cross-modal temporal biases is that they are shaped during a sensitive period of development. Sensitive periods are limited periods during brain development in which the influence of sensory experience on the brain is particularly strong (**Knudsen, 2004**). During a sensitive period, the architecture of a neural circuit is shaped to meet an individual's environment. The long-term result is a preference of the neural circuits for certain states of activity even if the environment dramatically changes (**Röder et al., 2007**; **Röder et al., 2004**; **Sourav et al., 2019**), without prohibiting short-term changes in the neural circuits' activity patterns (**Knudsen, 2004**; **Knudsen, 2002**). The role of early sensory experience for the genesis of perceptual biases is extremely difficult to address in humans; only individuals whose early sensory environments were atypical due to natural causes open a window into the role of experience in perceptual development. We tested the hypothesis of a sensitive period for cross-modal perceptual biases by asking individuals born with dense, bilateral cataracts whose sight was restored 6 to 168 months after birth (**Table 1**) to temporally order spatially separated events across vision, audition and touch.

## Results

Thirteen individuals with a history of congenital bilateral, dense cataracts which had been surgically removed (CC-group) participated in two spatial temporal order judgement tasks, ten in a visual-auditory task (Expt. 1) and ten in a visual-tactile task (Expt. 2). To test for the role of vision during infancy as well as to control for effects related to having had eye surgery and persisting visual impairments,

**Table 1.** Sample characteristics.

| | N | Sex | Handedness | Visual acuity of the better eye | Age at testing | Age at surgery | Time period between surgery and testing |
|---|---|---|---|---|---|---|---|
| CC – Expt. 1 visual-auditory | 10 | 8 males | 10 right-handed | 0.16–1.3 logMAR, mean 0.84 logMAR | 9–46 years, mean 30 years | 6–168 months, mean 60 months | 24–528 months, mean 296 months |
| DC – Expt. 1 visual-auditory | 9 | 7 males | 9 right-handed | −0.5–0.8 logMAR, mean 0.25 logMAR | 8–19 years, mean 13 years | 74–183 months, mean 120 months | 12–66 months, mean 30 months |
| CC – Expt. 2 visual-tactile | 10 | 10 males | 10 right-handed | 0.16–1.3 logMAR, mean 0.75 logMAR | 11–45 years, mean 30 years | 5–216 months, mean 57 months | 93–516 months, mean 296 months |
| DC – Expt. 2 visual-tactile | 9 | 6 males | 9 right-handed | 0–0.5 logMAR, mean 0.22 logMAR | 9–19 years, mean 14 years | 30–183 months, mean 102 months | 13–174 months, mean 63 months |

sixteen individuals, nine per experiment, who had undergone surgery for cataracts which had developed during childhood served as controls (DC-group). As both cataract-reversal groups differed in age (*Table 1*) and age is known to influence cross-modal temporal perception (*Röder et al., 2013*; *Noel et al., 2016*), their performance was not directly compared but each group was contrasted separately against age-matched typically sighted individuals (MCC- and MDC-groups). In every trial, two successive stimuli were presented, one in each hemifield (*Figure 1A*). Stimulus order and the time interval between the two stimuli (stimulus onset asynchrony; SOA) varied randomly across trials. Visual-auditory and visual-tactile stimulus pairs were randomly interleaved with unimodal stimulus pairs. Participants reported the side of the first stimulus irrespective of its modality (*Zampini et al., 2003*). For the CC-group, we predicted that due to their history of visual loss and persisting visual impairments participants would give auditory and tactile stimuli a higher preference than visual stimuli. Such a preference would result in a shift of the point of perceived simultaneity (PSS) –the SOA at which auditory-visual or tactile-visual pairs are perceived as simultaneous– towards SOAs with greater auditory or tactile lags, and an overall lower proportion of 'visual first'-responses compared to their controls. Additionally, we predicted a lower visual spatio-temporal resolution resulting in a lower proportion of correct temporal order judgments for the CC-group.

Most strikingly and contrary to our predictions, CC-individuals showed a bias toward perceiving visual stimuli as occurring earlier than auditory (*Figure 1D,E*; CC-group, Expt. 1, PSS, $t(9) = -1.51$, p=0.033, $r_{equiv} = 0.57$; 'visual first'-response probability, $\chi^2 (1)=4.31$, p=0.038, $r_{equiv} = 0.55$; see *Supplementary file 1* for full statistical models) and tactile stimuli (CC-group, Expt. 2, PSS, $t(9) = -3.04$, p=0.009, $r_{equiv} = 0.69$; 'visual first'-response probability, $\chi^2(1)=12.46$, p<0.001, $r_{equiv} = 0.85$), respectively. CC-individuals' bias toward perceiving visual stimuli as earlier stood in contrast to the bias observed for their matched controls (CC- vs. MCC-group, Expt. 1, visual-auditory, PSS, $t(18) = -2.34$, p=0.005, $r_{equiv} = 0.56$; 'visual first'-response probability, $\chi^2 (1)=8.48$, p=0.004, $r_{equiv} = 0.57$; Expt. 2, visual-tactile, PSS, $t(18) = -2.47$, p=0.008, $r_{equiv} = 0.53$; 'visual first'-response probability, $\chi^2 (1)=5.31$, p=0.021, $r_{equiv} = 0.45$) who perceived auditory stimuli as occurring earlier than visual stimuli (MCC-group, Expt. 1, visual-auditory, PSS, $t(9) = 2.30$, p=0.009, $r_{equiv} = 0.69$; 'visual first'-response probability, $\chi^2 (1)=4.18$, p=0.041, $r_{equiv} = 0.55$) and did not show a significant bias for visual-tactile comparisons (MCC-group, Expt. 2, visual-tactile, PSS, $t(9) = -0.48$, p=0.321, $r_{equiv} = 0.16$; 'visual first'-response probability, $\chi^2 (1)=0.15$, p=0.701, $r_{equiv} = 0.17$). In contrast to the CC-group, DC-individuals showed the typical bias toward perceiving visual stimuli as occurring later than auditory and tactile stimuli (DC-group, Expt. 1, visual-auditory, PSS, $t(8) = 1.42$, p=0.017, $r_{equiv} = 0.67$; 'visual first'-response probability, $\chi^2 (1)=7.14$, p=0.008, $r_{equiv} = 0.74$; Expt. 2, visual-tactile, PSS, $t(8) = 2.31$, p=0.017, $r_{equiv} = 0.67$; 'visual first'-response probability, $\chi^2 (1)=8.86$, p=0.003, $r_{equiv} = 0.79$). In fact, their spatio-temporal bias toward perceiving visual stimuli as later than tactile stimuli was stronger than the bias of their matched controls (DC- vs. MDC-group, Expt. 1, visual-auditory, PSS, $t(18) = 1.40$, p=0.123, $r_{equiv} = 0.29$; 'visual first'-response probability, $\chi^2 (1)=1.84$, p=0.174, $r_{equiv} = 0.23$; Expt. 2, visual-tactile, PSS, $t(18) = 2.17$, p=0.019, $r_{equiv} = 0.49$; 'visual first'-response probability, $\chi^2 (1)=8.08$, p=0.004, $r_{equiv} = 0.60$; MDC-group, Expt. 1, visual-auditory, PSS, $t(8) = 1.32$, p=0.152, $r_{equiv} = 0.36$; 'visual first'-response probability, $\chi^2 (1)=0.83$, p=0.362, $r_{equiv} = 0.13$; Expt. 2, visual-tactile, PSS, $t(8) = -0.36$, p=0.361, $r_{equiv} = 0.13$; 'visual first'-response probability, $\chi^2=0.85$, p=0.356, $r_{equiv} = 0.15$).

CC-individuals' proportion of correct temporal order judgments for visual and cross-modal stimulus pairs (*Figure 1F*) was reduced compared to their controls' temporal resolution (CC-group vs. MCC-group, Expt. 1, visual, $\chi^2 (1)=9.68$, p=0.002, $r_{equiv} = 0.62$; Expt. 1, visual-auditory, $\chi^2 (1)=17.88$, p<0.001, $r_{equiv} = 0.78$; Expt. 2, visual, $\chi^2 (1)=4.72$, p=0.030, $r_{equiv} = 0.42$; Expt. 2, visual-tactile, $\chi^2 (1) =4.03$, p=0.045, $r_{equiv} = 0.39$; see *Supplementary file 1* for full statistical models), but no significant difference between the CC-group and their controls emerged for auditory and tactile stimulus pairs, (CC-group vs. MCC-group, Expt. 1, auditory, $\chi^2 (1)=1.42$, p=0.233, $r_{equiv} = 0.17$; Expt. 2, tactile, $\chi^2 (1)=0.11$, p=0.742, $r_{equiv} = 0.15$). DC-individuals' spatio-temporal resolution was lower than the resolution of their controls independent of modality condition (DC-group vs. MDC-group, Expt. 1, all modalities, $\chi^2 (1)=4.98$, p=0.026, $r_{equiv} = 0.46$; Expt. 2, all modalities, $\chi^2 (1)=11.99$, p=0.001, $r_{equiv} = 0.68$).

Across participants, the size of the bias in the proportion of 'visual first'-responses in bimodal trials correlated negatively with the proportion of correct temporal order judgements in these trials (*Figure 1G*; Expt. 1, visual-auditory, r = −0.39, p=0.018; Expt. 2, visual-tactile, r = −0.49, p=0.002).

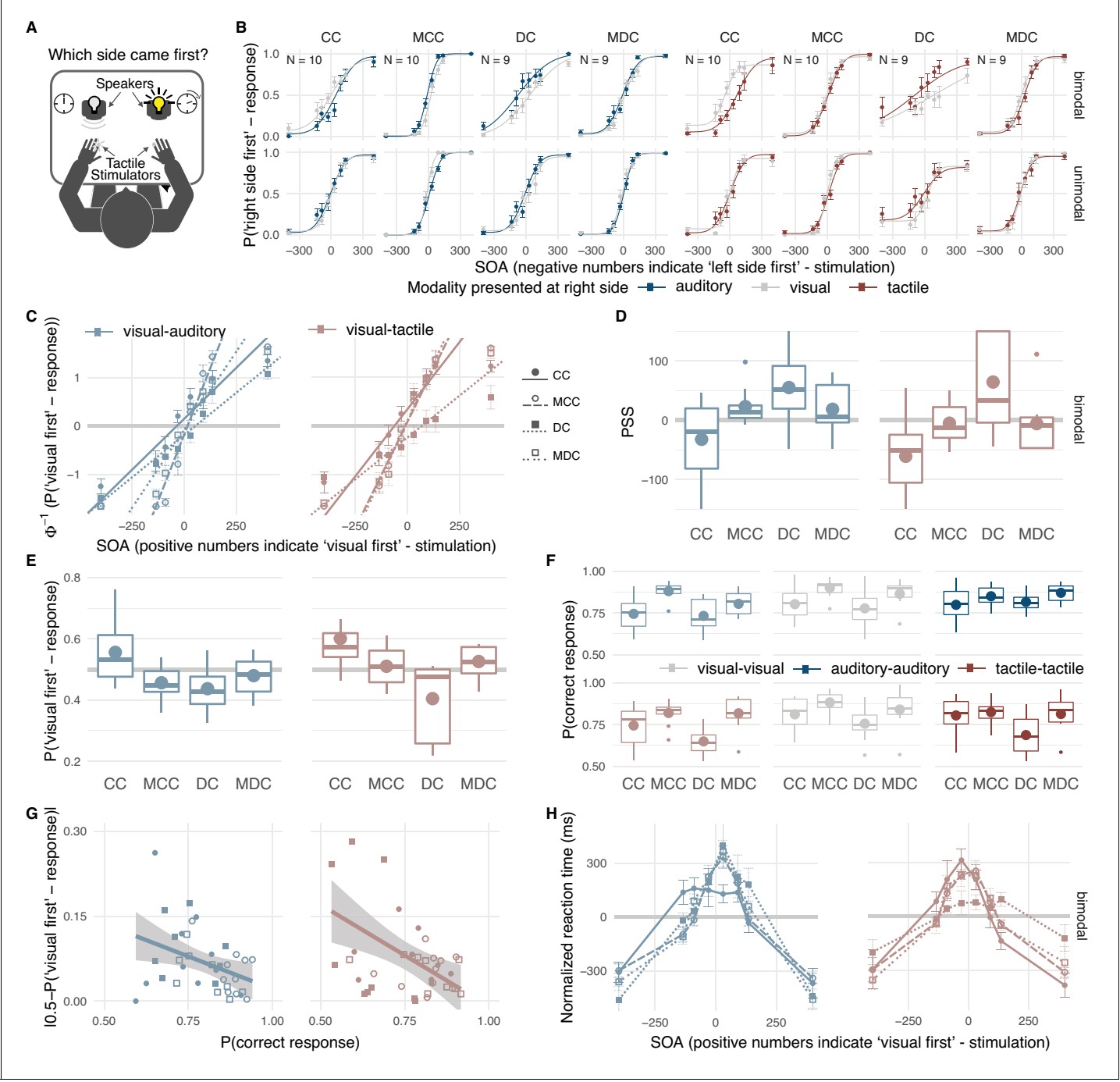

**Figure 1.** Effects of early visual experience on spatio-temporal order biases and resolution within and across vision, audition, and touch. Thirteen individuals with surgically removed congenital, bilateral dense cataracts (CC) and sixteen individuals whose reversed cataracts had developed later in life (DC) as well as normally sighted individuals matched for age, gender, and handedness (MCC, MDC) took part in the study (see *Table 1* for details about the samples). (**A**) Participants judged the spatio-temporal order of two successive stimuli –one presented in each hemifield– by indicating the location of the first stimulus. In Expt. 1, visual-visual, auditory-auditory, and visual-auditory stimulus pairs were presented, in Expt. 2, visual-visual, tactile-tactile, and visual-tactile stimulus pairs. (**B**) Psychometric curves: group average probabilities of a 'right side first'-response as a function of SOA (negative values indicate 'left side first'-stimulation). Data are split by the modality presented at the right side (visual = grey, auditory = dark blue, tactile = dark red) and the modality at the other side (rows, bimodal = different modality, unimodal = same modality). Curves display cumulative Gaussian distributions fitted to the group data for illustrative purposes. Error bars indicate standard errors of the mean in all panels. (**C**) Probit analysis: Group mean probits (CC = filled circle, MCC = open circle, DC = filled square, MDC = open square) of the probability for a 'visual first'-response as a function of SOA (positive values indicate 'visual first'-stimulation) in bimodal trials (visual-auditory = light blue, visual-tactile = light red). Lines show

*Figure 1 continued on next page*

*Figure 1 continued*

group averages of linear regression lines fitted to individual participant's data. The longest SOA was excluded from the linear regression, if a participant's performance had reached an asymptote. (D) Point of cross-modal subjective simultaneity (PSS): Box-and-whisker plots show the distribution (median, quartiles, minimum and maximum bounded at 1.5 x interquartile range, outliers) of individual PSS values; big circles indicate group mean values. Positive PSS values indicate that stimulus pairs in which the visual stimulus was presented first were perceived as simultaneous. (E) 'Visual first'-bias: Box-and-whisker plots show the distribution of the proportion of 'visual first'-responses per participant; big circles indicate group mean values. All SOAs were presented equally often, thus, a PSS of zero and a proportion of 'visual first'-responses equal to 0.5 would have been correct. A positive PSS corresponds to a proportion of 'visual first'-responses below 0.5 and vice versa. (F) Temporal order judgment accuracy: Distribution and group average probabilities of correct responses separately for each group and modality condition for each of three different modality combinations per experiment (light blue: visual-auditory, light red: visual-tactile, grey: visual-visual, dark blue: auditory-auditory, dark red: tactile-tactile). (G) Bias-accuracy relation: Single participants' proportion of correct responses (x-axis) plotted against the absolute value of their 'visual first'-bias (y-axis; markers as in (C)). Lines show linear regressions and 95% confidence intervals. (H) Reaction time (RT) distributions: Individually normalized RT in bimodal trials as a function of SOA (positive values indicate 'visual first'-stimulation) per group. Markers and lines show group mean values (markers and line styles as in (C)).

The online version of this article includes the following figure supplement(s) for figure 1:

**Figure supplement 1.** Excluded data.
**Figure supplement 2.** Visual-auditory temporal order judgments in typically sighted individuals in an additional experiment controlling for the effects of visual stimulus brightness.
**Figure supplement 3.** Just noticeable differences (JND).

The distribution of reaction times in bimodal trials peaked at different SOAs across groups (*Figure 1H*; CC- vs. MCC-group, Expt. 1, visual-auditory, $t(16) = 2.11$, p=0.041, $r_{equiv} = 0.40$; Expt. 2, visual-tactile, $t(18) = 1.56$, p=0.067, $r_{equiv} = 0.35$) with peaks being shifted in the same direction as the PSS.

## Discussion

Here, we investigated whether biases in cross-modal temporal perception are shaped during a sensitive period in early development by testing sight-recovery individuals with a history of congenital loss of pattern vision. In two spatio-temporal order judgement tasks, individuals with reversed congenital cataracts disproportionally often reported visual stimuli as occurring earlier than auditory and tactile stimuli (*Figure 1B–E*), exhibiting a reversed cross-modal bias compared to typically sighted controls and sight-recovery individuals whose cataracts had developed during childhood. These results, for the first time, demonstrate a sensitive period for cross-modal temporal biases. Moreover, the ability to correctly determine the spatio-temporal order of separate events across different sensory modalities and within vision was reduced after a transient phase of visual loss both after birth and later in childhood (*Figure 1F*) suggesting a persistent impairment of visual and cross-modal temporal resolution following transient visual deprivation.

At first glance, CC-individuals' bias to perceive visual events as earlier than auditory and tactile events seems to indicate that visual stimuli were processed faster than auditory and tactile stimuli following transient, congenital visual deprivation. However, CC-individuals' response accuracy (*Figure 1F*) indicated temporal processing impairments for vision but not for audition and touch. Consistently, studies measuring event-related potentials have provided no evidence for a shorter latency of the first visual cortical response in CC-individuals (*Sourav et al., 2018*; *Mondloch et al., 2013*; *McCulloch and Skarf, 1994*) and behavioral studies have demonstrated no advantage in reaction times to simple visual stimuli (*Putzar et al., 2012*; *de Heering et al., 2016*). Moreover, reduced visual contrast – typical for cataract-reversal individuals – is associated with delayed responses of the visual system and lower visual temporal sensitivity (*Watson, 1986*; *Stromeyer and Martini, 2003*). In sum, there is currently no evidence indicating accelerated processing of visual information following congenital visual deprivation.

During sensitive periods, experience shapes neural circuits customizing them to an individual's body and environment (*Knudsen, 2004*). CC-individuals' reversed cross-modal bias towards perceiving visual stimuli as occurring earlier than auditory and tactile stimuli might be rooted in consistent exposure to lagging visual input during an early sensitive period. Residual light perception, which exists even in the presence of dense cataracts, is typically sluggish and retinal transduction rates have been reported to be reduced in patients with dense, untreated cataracts (*Yamauchi et al.,*

2016). Moreover, a suppression of visual cortex activity has recently been observed in CC-individuals in the context of cross-modal stimulation (*Guerreiro et al., 2016*). Thus, CC-individuals likely were exposed to consistently delayed visual signals (with respect to auditory and tactile signals) before the cataracts were removed. Recalibration studies have shown that exposure to a consistent visual delay results in a bias towards perceiving visual input as earlier (*Fujisaki et al., 2004*; *Vroomen et al., 2004*), consistent with the present results in the CC-group. Thus, we suggest that the reversed bias exhibited by CC-individuals reflects adaptation to their atypical sensory environment during infancy, which results in structural differences in the neural circuits processing temporal order across modalities.

Individual cross-modal temporal biases in typically sighted humans might reflect the relative timing of sensory information across modalities during infancy, too. One-month old infants have been found to already show rudimentary cross-modal temporal biases for audition and vision (*Lewkowicz, 1996*). Slight differences in the relative rate of brain development across individuals, for example, in the myelination of neural circuits in different sensory areas, could give rise to the characteristic but stable (*Sanford, 1888*; *Grabot and van Wassenhove, 2017*) inter-individual differences in cross-modal temporal biases in humans. In turn, the variation of individual cross-modal temporal biases as a function of task (*Freeman et al., 2013*; *Ipser et al., 2018*; *Ipser et al., 2017*) might arise from differences in the relative rate of development across cortical areas. In sum, we suggest that individual cross-modal temporal biases (*Zampini et al., 2003*; *Grabot and van Wassenhove, 2017*) show a high stability because the brain had optimized cross-modal temporal perception (*Roseboom et al., 2015*) during a sensitive period which constitutes a setpoint for all future recalibration.

Additionally, the congenital absence of pattern vision might have increased CC-individuals' attention towards this modality after sight restoration, resulting in a permanent prior entry effect for vision (*Titchener, 1908*; *Spence et al., 2001*; *Vibell et al., 2007*; *Shore et al., 2001*). However, a previous study found longer switch times from audition to vision but not from vision to audition in CC-individuals whose cataracts had been removed in the first months of life than in controls, which was interpreted as indicating an attentional bias towards audition over vision (*de Heering et al., 2016*). Moreover, an attentional bias towards vision would have been expected to result in relatively shorter visual reaction times and early cortical potentials, which, as argued at the beginning of the Discussion is incompatible with the literature (*Sourav et al., 2018*; *Mondloch et al., 2013*; *McCulloch and Skarf, 1994*; *Putzar et al., 2012*; *de Heering et al., 2016*). Thus, while scarce existing data points towards a reduced dominance of vision in CC-individuals, the consequences of congenital loss of pattern vision for modality-specific attention are not yet understood.

The finding, that CC-individuals but not DC-individuals showed a reversed visual-auditory and visual-tactile temporal bias strongly suggests that perceptual biases are shaped by sensory experience during the initial period of life. In contrast to CC-individuals, DC-individuals had encountered unimpaired cross-modal stimulation after birth, which we suggest had enabled them to establish a typical bias. An early sensitive period would have prevented DC-individuals –who lost their vision later– from adapting this typical bias to their changed sensory environment. As a consequence, their partially severe, persisting visual impairments might have caused an increase of the typical bias of perceiving vision as delayed. However, the bias of some DC-individuals seems too large to be accounted for by delays in visual processing alone. The very high temporal uncertainty of some DC-individuals might have resulted in an overestimation of their cross-modal biases (*Figure 1G*): In agreement with Bayes' law, temporal perceptual judgments tend to rely more on prior information such as existing biases when the sensory information is uncertain (*Shi and Burr, 2016*; *Vercillo et al., 2015*). Consistent with this idea, the current data indicated a correlation between the size of the bias and the degree of sensory uncertainty. The sensitive period for cross-modal temporal perception might culminate during the first six months of life, the minimal duration of deprived vision in our CC-group. A previous study tested CC-individuals whose vision was restored within early infancy (4 months of age on average) in an audio-visual simultaneity judgment task with spatially aligned visual-auditory and visual-tactile stimulus pairs and did not report a reversed cross-modal bias (*Chen et al., 2017*). However, simultaneity judgments of spatially aligned cross-modal stimuli are simpler and less sensitive to biases than the spatial temporal order judgements we used (*van Eijk et al., 2008*; *Linares and Holcombe, 2014*). Moreover, both tasks rely on different neuronal processes (*Basharat et al., 2018*) and even one-month old infants seem to be able to evaluate

simultaneity (*Lewkowicz, 1996*). Thus, it is questionable whether the previous and our experimental design addressed the same multisensory processes. In sum, our results provide strong evidence for a sensitive period for cross-modal temporal perception, which leads to the consolidation of cross-modal spatio-temporal biases during early human ontogeny.

Typically, reaction times increase with increasing perceptual uncertainty. In temporal order judgements, uncertainty is highest for simultaneously perceived cross-modal stimuli. Thus, reaction times are expected to be longest at the PSS, if the PSS reflects biases in the perceived relative timing of the modalities, but not if the PSS reflects a response bias towards one modality (*Spence, 2010*; *Shore et al., 2001*). Here, CC-individuals' reaction times and thus sensory uncertainty peaked when the visual stimulus slightly lagged behind the auditory or the tactile stimulus (*Figure 1H*), whereas all other participants responded slowest when the visual stimulus led by a short amount of time. Thus, even though response speed was not stressed in the instructions, reaction times of the present study suggest that the observed biases reflect perceptual shifts in the relative timing of cross-modal sensory information.

Both cataract-reversal groups exhibited lower spatio-temporal accuracy (*Figure 1F*) –indicative of a decreased temporal resolution and an increased lapse rate (*Figure 1B*)– than their controls in the visual and the cross-modal tasks. Persisting visual impairments might have contributed to the reduced visual spatio-temporal resolution of both cataract-reversal groups given that visual contrast affects visual temporal perception (*Watson, 1986*; *Stromeyer and Martini, 2003*). The observation that a lower temporal resolution was found not only for visual but additionally for cross-modal temporal order judgments suggests a strong dependence of cross-modal temporal ordering on the visual sense. This might be related to the spatial nature of our task; training of non-spatial, purely visual temporal order judgments did not lead to an improved visual-auditory temporal resolution (*Alais and Cass, 2010*). The finding that both cataract groups exhibited a lower temporal resolution but only the CC-group an altered cross-modal bias strongly suggests that cross-modal spatio-temporal biases and resolution are dissociable processes. The conjunction of CC- and DC-individuals' reduced resolution might point towards a long sensitive period for the development of spatio-temporal sensory resolution which would be compatible with the protracted developmental time course of cross-modal temporal ordering of spatially separate events (*Röder et al., 2013*).

Furthermore, the present finding of increased cross-modal temporal uncertainty could explain why recent studies have found altered multisensory integration for some but not all functions following congenital, transient periods of visual deprivation (*Chen et al., 2017*; *Putzar et al., 2007*; *Guerreiro et al., 2015*). An increased spatio-temporal uncertainty hinders the detection of temporal correlations (*Parise and Ernst, 2016*) between complex signals such as speech stimuli (*Putzar et al., 2007*; *Guerreiro et al., 2015*). At the same time, higher temporal uncertainty predicts wider temporal integration windows for simple, spatially-aligned stimuli (*Chen et al., 2017*), which in turn might have enabled typical multisensory redundancy gains for such cross-modal stimuli (*Putzar et al., 2012*; *de Heering et al., 2016*). It is interesting to note, that this pattern mirrors the typical development of multisensory temporal perception. Very young infants integrate simple visual and auditory stimuli across a wide window of asynchronies but do not integrate more complex stimuli based on other features than stimulus onset (*Lewkowicz, 1996*; *Lewkowicz, 2012*). The multisensory integration window narrows during childhood (*Noel et al., 2016*; *Lewkowicz and Flom, 2014*; *Hillock et al., 2011*), in parallel to the improvement of unisensory temporal perception (*Brannon et al., 2007*). Some developmental theories suggest that more complex temporal multisensory functions build on previously acquired temporal multisensory skills (*Lewkowicz, 2000*) predicting that the impact of transient loss of pattern vision on multisensory perception should as suggested for the visual system (*Hyvärinen et al., 1981*; *Maurer et al., 2005*), increase with increasing task difficulty.

In conclusion, congenital but not late transient visual deprivation was associated with a bias towards perceiving visual events as earlier than auditory or tactile events in sight-recovery individuals, suggesting that cross-modal temporal biases depend on sensory experiences during an early sensitive period.

## Materials and methods

### Participants

The sample of the visual-auditory experiment (Expt. 1) comprised ten individuals who were born with bilateral dense cataracts (congenital cataracts, CC) and whose vision was restored later in life (for details see *Table 1*) and nine individuals with transient, bilateral cataracts which had developed during childhood (developmental cataracts, DC). The sample tested in the visual-tactile experiment (Expt. 2) comprised ten CC- and nine DC-individuals. For each CC- and DC-participant an age-, gender- and handedness-matched control participant was recruited. Seven CC- and two DC-individuals as well as 13 control participants took part in both experiments. The majority of participants with a history of cataracts were recruited and tested at the LV Prasad Eye Institute in Hyderabad, India. Three CC-individuals and all control participants were recruited and tested at the University of Hamburg, Germany. The presence of congenital cataracts was affirmed through an analysis of the medical records by the participant's optometrist and ophthalmologists. Since cataracts were sometimes diagnosed at a progressed age, additional criteria such as presence of nystagmus, strabismus, the density of the lenticular opacity, the lack of fundus visibility prior to surgery, a family history of congenital cataracts, and parents' reports were employed to confirm the onset of the cataract. Data of five additional participants were excluded from all analyses because (a) the onset of the cataract remained unclear (two participants, one took part in the visual-auditory experiment and one in the visual-tactile experiment), (b) the time period between surgery and testing was shorter than 12 months (one participant from the CC-group who took part in both experiments and one participant from the DC-group who took part in the visual-auditory experiment), or because (c) additional neurological problems were suggested by medical records (one participant from the CC-group who took part in both experiments). Data of two further participants (one CC- and one DC-individual) were excluded because they performed even in the easiest 400-ms-long SOA condition at chance level or below in the visual-auditory experiment. The 400-ms-long SOA was included to check whether participants had understood the task, which was explained to them by an interpreter. All excluded data are shown in the supplementary information (*Figure 1—figure supplement 1*) and are in accordance with the results presented in the main text. Adult participants and legal guardians of minors were reimbursed for travel expenses, accommodation, and absence from work, if applicable; adult participants tested in Hamburg received a small monetary compensation or course credit. Children received a small present. All participants or, if applicable, their legal guardian, provided written informed consent before beginning the experiment. The study was conducted in accordance with the ethical guidelines of the Declaration of Helsinki and was approved by the ethical board of the German Psychological Society as well as the local ethical committee of the LV Prasad Eye Institute.

### Apparatus and stimuli

Participants sat at a table, facing two speakers, positioned at 14° visual angle (15 cm at 60 cm distance) to the left and to the right of the participant's midline (*Figure 1A*). Three LEDs were mounted on top of each speaker. In the visual-tactile experiment, custom-made, noise-attenuated tactile stimulators were attached to the dorsal sides of both index fingers. A stimulus lasted 15 ms, independent of modality. During a stimulus, the LEDs emitted red light, the speakers played white noise, and the tactile stimulators vibrated at a frequency of 100 Hz. All three LEDs were used for cataract-reversal participants, but only one LED for typically sighted participants, to roughly compensate for persistent visual impairments in cataract-reversal participants. To rule out that typically sighted participants perceived vision as delayed due to the lower number of LEDs, we tested five additional typically sighted participants (all female and right-handed, 23–50 years old, mean age 34 years) in the visual-auditory experiment while using all three LEDs. These participants showed a significant typical bias towards perceiving vision as delayed ($t(4)=6.36$, $p=0.003$, *Figure 1—figure supplement 2*). Constant white noise was presented from a centrally located speaker, to mask residual noise produced by the tactile stimulators. During the experiment, participants fixated a mark placed centrally between the loudspeakers and rested both hands on buttons aligned with the loudspeakers (visual-auditory experiment) or both feet on foot pedals (visual-tactile experiment). Younger participants sometimes experienced problems activating the response devices in a controlled manner. These participants (visual-auditory experiment: one CC- individual; visual-tactile experiment: two DC-

individuals) and their controls responded by waving one hand and the experimenter entered the response. The experiment was controlled by Presentation (Version 17.1.05, Neurobehavioral Systems, Inc, Berkeley, CA, www.neurobs.com), which recorded responses and interfaced with custom-built hardware to drive the stimulators.

## Task, procedure, and design

In each trial, two stimuli were presented in close succession; one stimulus in each hemifield. Participants indicated at which side they perceived the first stimulus. Responses had to be withheld until the second stimulus had been presented. Response times were not restricted, and the next trial started 2 s after the response had been registered.

The modality of the stimulus presented at either side (visual or auditory, Expt. 1; visual or tactile, Expt. 2) and the stimulus onset asynchrony (SOA;±30,±90,±135,±400 ms, with negative SOAs indicating 'left side first'-stimulus pairs) of the two stimuli varied pseudo-randomly across trials. Each of the 32 stimulus conditions (2 modalities x 2 sides x 8 SOAs) was repeated 10 times; the 320 trials were divided into 10 blocks. Participants additionally completed ten practice trials with an SOA of ±400 ms at the beginning of the experiment. If necessary, the practice trials were repeated until participants felt confident about the task. In the visual-tactile experiment, a subsample of participants was additionally tested while holding the hands crossed (data not reported here). Participants were encouraged to take breaks in between blocks. Some of the cataract-reversal participants did not complete the full experiment, mostly due to time constraints. Except for practice trials, participants did not receive feedback.

## Data analysis

Data and analysis scripts are made available online (*Badde et al., 2019*). Trials with reaction times shorter than 100 ms and more than 2.5 standard deviations above the participant's mean reaction time (RT) were excluded from the analysis (2.1% of trials; responses entered by the experimenter were not filtered).

Each participants' data were split according to the modality of the stimulus pair (visual-visual, auditory-auditory, or visual-auditory for Expt. 1; visual-visual, tactile-tactile, or visual-tactile for Expt. 2). Participants' left-right responses were transformed (a) into binary 'right first' – values (1 = 'right first'-response, 0 = 'left first'-response; *Figure 1B*), and (b) for bimodal trials only, into binary 'visual first' – values (1 = 'visual first'-response, 0 = 'auditory/tactile first'-response).

To test for temporal order biases toward one modality, two complementary analyses of 'visual first'-values were conducted. (1) We estimated each participant's point of subjective simultaneity (PSS), the theoretical SOA at which a visual and a non-visual stimulus are perceived as simultaneous. To this aim, we linearized the proportion of 'visual first'-responses as a function of SOA (with positive values indicating 'visual first'-stimulation) using a probit transformation and fitted a linear regression line (*Figure 1C*). If a participant's performance had reached an asymptote, that is if performance at the longest SOA was equal to that at the second longest SOA, the longer SOA was dropped to allow for a better fit. The PSS equals the zero point of the linear function. PSS estimates were bounded at ±150 ms; the results pattern was not influenced by the value of the boundary (*Figure 1D*). To test for group effects, we conducted a one-way ANOVA and followed-up by planned unpaired *t*-tests comparing the CC- and the DC-group each with their respective control groups. Moreover, each groups' PSS values were tested against zero with a *t*-test. As the normality assumption was violated, permutation tests were used to derive a null-distribution and thus non-parametric *p*-values for all tests. (2) We additionally analyzed participants' probability of a 'visual first'-response independent of stimulation, that is across SOAs. This measure corresponds to the criterion in signal detection theory and was added because it is robust against outliers along the psychometric functions. Single trial 'visual first'-values were fitted with a generalized linear mixed model with a binomial distribution family and a log link function. Group was included as fixed effect and participants were treated as random effects. As planned comparisons, we first calculated pairwise contrasts comparing each cataract-reversal group with its matched control group and second estimated fixed contrasts separately for each group to evaluate whether the probability to perceive the visual stimulus before the auditory or tactile stimulus significantly differed from chance level.

To analyze the spatio-temporal resolution across groups and modality conditions, single trial accuracy values (1 = correct, 0 = incorrect) were fitted with a generalized linear mixed model with a binomial distribution family and a log link function. Group and stimulus modality were included as fixed effects and participants were again included as random effects. To resolve interactions between both predictors, we first conducted pairwise contrasts on both predictors comparing group differences between each cataract-reversal group and its matched control group across modalities and second pairwise contrasts testing for group differences separately for each modality condition. Additionally, we derived and report another measure of temporal resolution, just noticeable difference (JND) values, that is the SOA at which participants perceive the correct temporal order with 75% probability (*Figure 1—figure supplement 3*). However, for participants whose performance had not reached an asymptote at an SOA of ±135 ms these estimates are less reliable than for participants for whom this was the case. The first situation was more frequent in both the CC- and the DC-group, while the second situation was more frequent in the control groups. Therefore, we refrained from statistical group comparisons for JNDs.

To explore a possible relation between participant's cross-modal temporal bias and temporal resolution, a Pearson correlation was calculated across participants from all groups. The size of the bias was defined as the distance between the proportion of 'visual first'-responses and 0.5, which is equivalent to no bias. The resolution was defined as the proportion of correct temporal order judgments. To avoid leverage effects, data points exceeding 2.5 standard deviations distance from the group mean in any measure were excluded.

Finally, reaction times (RT) in bimodal trials were analyzed as a function of SOA (with positive values indicating 'visual first'-stimulation). We normalized each participant's RT (responses entered by the experimenter were excluded) across SOAs and compared maximum likelihood estimates of the SOA at which the RT distribution peaked across groups. Permutation tests of the unpaired *t*-statistic were employed for the comparisons, because the normality assumption was violated.

We report $r_{equivalent}$ (*Rosenthal and Rubin, 2003*) as effect size estimate, since no generally accepted standardized effect size measure exists for mixed models and permutation analyses.

## Acknowledgements

Our work was supported by the European Research Council (ERC-2009-AdG 249425 CriticalBrain-Changes) to BR and the German Research Foundation (DFG) with a research fellowship grant to SB (BA 5600/1–1) and DFG Ro 2625/10–1 to BR as well as by the University of Hamburg with a post-doctoral fellowship to SB. We thank Marlene Hense, Nicola Kaczmareck, Carla Petroll, Lea Hornung, Deniz Froemke, and Rakesh Balachandar for help with data acquisition as well as Kabilan Pitchaimu-thu and Prativa Regmi for help with data curation. We thank D Balasubramanian of the L V Prasad Eye Institute for supporting this study.

## Additional information

### Funding

| Funder | Grant reference number | Author |
|---|---|---|
| Deutsche Forschungsge-meinschaft | BA 5600/1-1 | Stephanie Badde |
| H2020 European Research Council | ERC-2009-AdG 249425 CriticalBrainChanges | Brigitte Röder |
| Deutsche Forschungsge-meinschaft | DFG Ro 2625/10-1 | Brigitte Röder |
| University of Hamburg | postdoctoral fellowship | Stephanie Badde |

The funders had no role in study design, data collection and interpretation, or the decision to submit the work for publication.

## Author contributions
Stephanie Badde, Data curation, Software, Formal analysis, Supervision, Visualization, Methodology, Writing - original draft, Writing - review and editing; Pia Ley, Software, Investigation, Methodology, Writing - review and editing; Siddhart S Rajendran, Investigation, Writing - review and editing; Idris Shareef, Data curation, Writing - review and editing; Ramesh Kekunnaya, Resources, Investigation, Writing - review and editing; Brigitte Röder, Conceptualization, Resources, Supervision, Funding acquisition, Investigation, Methodology, Writing - original draft, Writing - review and editing

## Author ORCIDs
Stephanie Badde (iD) https://orcid.org/0000-0002-4005-5503
Idris Shareef (iD) http://orcid.org/0000-0001-9258-2199
Ramesh Kekunnaya (iD) http://orcid.org/0000-0001-5789-2300

## Ethics
Human subjects: All participants or, if applicable, their legal guardian, provided written informed consent before beginning the experiment. The study was conducted in accordance with the ethical guidelines of the Declaration of Helsinki and was approved by the ethical board of the German Psychological Society as well as the local ethical committee of the LV Prasad Eye Institute.

## Decision letter and Author response
Decision letter https://doi.org/10.7554/eLife.61238.sa1
Author response https://doi.org/10.7554/eLife.61238.sa2

# Additional files

## Supplementary files
• Supplementary file 1. Supplementary statistical information.
• Transparent reporting form

## Data availability
All data have been deposited on Open Science Framework under CQN48.

The following dataset was generated:

| Author(s) | Year | Dataset title | Dataset URL | Database and Identifier |
|---|---|---|---|---|
| Badde S, Ley P, Rajendran SS, Kekunnaya R, Shareef I, Röder B | 2019 | Cross-modal temporal biases emerge during early sensitive periods | https://doi.org/10.17605/OSF.IO/CQN48 | Open Science Framework, 10.17605/OSF.IO/CQN48 |

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
