## [Decision Letter]

**Acceptance summary:**

In this study the authors compare congenital and developmental cataract survivors to typically developed individuals, to study the influence of early life experience on multisensory processing. It is this double comparison that allows them to suggest that perceptual biases develop in early life. We were most excited about the inclusion of the late-cataract control group, as well as the rich dataset, which includes replication of the findings during visuo-tactile and visuo-auditory multisensory processing. We believe this unique dataset will set a new bar for retrospective behavioural research of sensitive periods.

**Decision letter after peer review:**

[Editors’ note: the authors submitted for reconsideration following the decision after peer review. What follows is the decision letter after the first round of review.]

Thank you for submitting your work entitled "Cross-modal temporal biases emerge during early sensitive periods" for consideration by *eLife*. Your article has been reviewed by three peer reviewers, including Tamar R Makin as the Reviewing Editor and Reviewer #1, and the evaluation has been overseen by a Senior Editor.

Our decision has been reached after consultation between the reviewers. Based on these discussions and the individual reviews below, we regret to inform you that your work, as currently presented, will not be considered further for publication in *eLife*.

All reviewers were very excited about the general experimental approach involving comparisons of congenital versus developmental cataract survivors for studying the influence of early life experience on multisensory processing. The reviewers agreed that the presented findings were intriguing. However, even when considering the unique challenges of data collection with these precious populations, it felt that the interpretation of the findings was overly simplistic. In particular, they felt that several alternative accounts need to be better considered in order to interpret these findings more mechanistically. It was agreed that further experimental evidence will need to be presented in order to tease apart these alternative accounts, and in particular the role of attention. If a more comprehensive account for this data could be offered, then we will be very happy to consider a resubmission of the manuscript. Please note that the reviewers expressed an interest in the alternatives provided for interpreting the results, so they do not necessarily expect that these will be discounted as confounds. Instead, they are looking for more conclusive evidence for interpreting the mechanisms driving the findings. We leave it to the authors to decide what type of evidence could be raised to best address the reviewers issues, be it new experiments acquired from control participants, more detailed analysis of the existing data or presentation of additional data collected with the cataract groups. If you wish to resubmit your work after substantial revisions, please provide a point-by-point response to address each of the reviewers major comments. Of course, we understand should you wish to take this present version elsewhere at this time.

Reviewer #1:

I think it’s a very exciting dataset and the results are quite interesting. I also liked the main interpretation of the results. But at present, this all feels a little too speculative/tentative. I feel like important opportunities in analysis of this precious data have been missed, and that at present we are only provided a very partial glims at the results.

1) I'm not entirely clear about the choices for the statistical analysis, particularly with respect to the “bias” parameter. To begin with, why did they chose a hierarchical logistic regression? Why is it advantageous to, say, signal detection analysis which is often used to study biases? It is also not clear how the ISI parameter is handled in the model

2) considering the relative richness of the data collected, the analysis is quite simplistic and leaves something to be desired. Why did they not follow standard analysis procedures based on the ISI in their design (i.e. psychometric function, calculation of the slope/JND and point of subjective equality)? Considering the tendency to report “visual first” could be modulated by several dissociated factors (as elaborated in the Discussion), I feel like the traditional parameters could help interpret the described bias better. For example, attentional effects could be hypothesised to be impacting all ISI similarly (a shift of the curve), whereas the experience-based structural differences should be more time-dependent, based on biological onset asynchrony (i.e. shifting the slope)?

3) Perhaps the most interesting comparison is between the DC and CC groups, especially considering methodological and technical differences between the cataract groups and the controls. But this wasn't reported in the results? From the figures it appears that biases were indeed different (?) but it also seems that they might sustain different visual and tactile abilities (Figure 1D)? How would that potentially impact the main results?

4) The MDC group didn’t show the typical visual bias effect, this is a risk when using small sample sizes, and thus requires some consideration. Was this group not a representative control group? Again, based on Figure 1C it indeed appears that their performance at the bimanual condition was low. I appreciate that the interpretation of the DC group stands alone, by showing a bias effect in the vision direction (and opposite from the CC group). But still, this should be considered. Please don't address this question by showing no significant differences between the MDC and MCC groups, unless there are clear evidence to support this as lack of a difference (e.g. bayesian statistics)

Reviewer #2:

Badde and colleagues presented a cross-modal temporal task (which stimulus was presented first) to controls, individuals born with dense, bilateral cataracts whose sight was restored after birth (CC), and those who developed cataracts after childbirth (DC). They found that those born with cataracts that were subsequently removed (the CC group) were more likely to respond visual 1st (vs. auditory or tactile) compared to the DC and matched control groups. The results are intriguing. However, there are major concerns with regards to the interpretation that diminish enthusiasm for these findings.

The authors claim that "cross-modal perceptual biases are not innate but rather acquired during a sensitive period." My primary question is whether the observed effects are truly "cross-modal perceptual biases". An alternative hypothesis is based on the "law of prior entry", in which "the object of attention comes to consciousness more quickly than the objects which we are not attending to" (Titchener, 1908). The authors are aware of this, citing articles that show biases in temporal order when attending unequally across modalities (Titchener, 1908; Shore, Spence and Klein, 2001; Spence, Shore and Klein, 2001; Vibell et al., 2007). They then argue that (a) "attention has been ruled out as a cause of persistent biases in cross-modal spatiotemporal processing" and (b) "it is not obvious why CC- and DC-individuals would entertain opposing attentional foci given that both groups experienced transient severe visual impairment and suffer from remaining visual acuity impairments." For (a), I am not sure how their citation has ruled out attention as a cause of these biases from their Abstract: "we showed that temporal-order perception may be considered a psychological bias that attention can modulate but not fully eradicate". For (b), the argument would be that those w/o high-quality visual information at birth (CC) would need to attend more to the visual stimulus due to deficits/differences in their visual systems at birth, whereas those in DC group wouldn't need to, given that they were born with intact visual input.

This is not to say that the author's hypothesis or this alternative hypothesis aren't interesting, but more data would be needed to tease these apart. This leads to my second issue, the paper is lacking in mechanism. The Introduction (especially for the lay reader) provides minimal information about why this bias occurs, and why this matters. The paper could be improved substantially with a more mechanistic rationale, especially in the Introduction.

Finally, the authors note that the visual first bias in the CC group did not significantly correlate with duration of visual deprivation in CC group. Is this meaningful given the small number of subjects tested? It seems like there isn't enough evidence, as presented, to make this claim.

Reviewer #3:

The authors investigate cross-modal temporal perceptual biases in participants born with congenital bilateral cataract and treated on different ages. They find that in this population, the biases are opposite to healthy and developmental cataract populations. They conclude such biases develop in early sensitive period. The paper is intriguing and the results are important, however, it is to my view that the interpretation of the results and the main conclusion might be problematic.

1) The authors main point is trying to rule out the possibility of innate cross-modal perceptual biases. The option of those biases being innate might still be possible, but biases might be reversed due to lack of visual input at early age while the brain is still flexible enough. This possibility should be mentioned. So, throughout the paper, I believe the claim of lack of innateness should be transform to that of sensitive period and innateness should only be discussed as a possibility in the Discussion. If there are any developmental studies showing the emergence of the biases during the first few months after birth, they should be mentioned since they can support the authors' claim. The word "emerge" at the heading should also be change to something more accurate.

2) In the Introduction: "Moreover, it is not obvious why CC- and DC-individuals would entertain opposing attentional foci given that both groups experienced transient severe visual impairment and suffer from remaining visual acuity impairments." The authors cannot use this claim to rule out attentional foci since it is also not obvious why CC and DC individuals will have an opposing cross-modal effect as well (this study demonstrates this exact result). Attention could be opposite for the same unknown reason. I can think of one reason why CC and DC groups will entertain opposing attentional foci. Developmental individuals were used to having high resolution input, now resolution decreased to medium so they are less attentive to vision. CCs were used to having low resolution input, now resolution increased to medium, so they are more attentive. This could be one possible reason why both groups will have opposing results in visual tests.

3) DC group differed from the control group in the visuo-tactile experiment (with a trend in the visuo-auditory). Doesn't it mean that the sensitive period for the cross-modal perceptual biases is very long? Since those biases were altered (increased, not reversed) at a relatively late age? At least, this difference should be acknowledged in the Discussion acknowledging that there is no theory justifying this effect.

4) Overall, following my previous comments, I think one can only conclude that cross-modal biases are sensitive to visual experience. If a stronger claim is needed, it should take into account both CC and DC groups' results.

---

## [Author Response]

[Editors’ note: the authors resubmitted a revised version of the paper for consideration. What follows is the authors’ response to the first round of review.]

Reviewer #1:I think it’s a very exciting dataset and the results are quite interesting. I also liked the main interpretation of the results. But at present, this all feels a little too speculative/tentative. I feel like important opportunities in analysis of this precious data have been missed, and that at present we are only provided a very partial glims at the results.1) I'm not entirely clear about the choices for the statistical analysis, particularly with respect to the “bias” parameter. To begin with, why did they chose a hierarchical logistic regression? Why is it advantageous to, say, signal detection analysis which is often used to study biases? It is also not clear how the ISI parameter is handled in the model

Our choice for an analysis with generalized linear mixed models was based on the sample size and composition. Even though large for this patient group, the sample size is small when compared to studies in healthy volunteers. And, though the participants were highly homogenous with respect to their visual history, they considerably varied in other parameters that are known to influence temporal order perception such as age. Mixed models are especially suited for such a situation because participants are included as random effects in the model. We have added this in the revised manuscript (subsection “Data Analysis”).

The bias parameter indicates the probability of a “visual first”-response across stimulus conditions. It is identical to the criterion in signal detection theory (c = (hit rate + false alarm rate)/2). In the revised manuscript, we now point this relation out (subsection “Data Analysis”).

The time interval between the stimuli was not included in this analysis because generalized mixed models require a considerable number of trials and this demand increases non-linearly with each additional factor. We now report an additional analysis based on the psychometric function (see below). Please note, an additional advantage of the original analysis is its robustness against outliers in the psychometric functions, which arise due to constraints in the number of trials that are inevitable in this type of research. (Several of the patients recruited in India were children. They simply cannot complete the same number of trials and with the same degree of concentration as university students who are used to computers or even psychological experiments). Thus, in the revised manuscript we report both analyses.

2) Considering the relative richness of the data collected, the analysis is quite simplistic and leaves something to be desired. Why did they not follow standard analysis procedures based on the ISI in their design (i.e. psychometric function, calculation of the slope/JND and point of subjective equality)? Considering the tendency to report “visual first” could be modulated by several dissociated factors (as elaborated in the Discussion), I feel like the traditional parameters could help interpret the described bias better. For example, attentional effects could be hypothesised to be impacting all ISI similarly (a shift of the curve), whereas the experience-based structural differences should be more time-dependent, based on biological onset asynchrony (i.e. shifting the slope)?

We have added a classical analysis of the point of subjective simultaneity; the results match those derived with the original bias measure, thus validating the latter.

The psychometric curves were shifted (Figure 1B); thus, the effect is of similar size across all short SOAs. This pattern is consistent with an auditory/tactile delay built into the neural circuits underlying temporal order judgments as well as with an attention-mediated visual processing gain following congenital loss of vision.

We analyzed both measures, the PSS and the original bias parameter, as the former is more precise but the latter more robust. In the revised manuscript, we moreover investigated the proportion correct and additionally report JNDs but only as a descriptive measure. Our JND estimates suffer from the fact that we sampled many short but only one long SOA, a design which was tailored to gather best-possible insights into the bias (subsection “Data Analysis”).

3) Perhaps the most interesting comparison is between the DC and CC groups, especially considering methodological and technical differences between the cataract groups and the controls. But this wasn't reported in the results? From the figures it appears that biases were indeed different (?) but it also seems that they might sustain different visual and tactile abilities (Figure 1D)? How would that potentially impact the main results?

We compared CC- and DC-groups each to their controls but not to each other because of the large age difference between groups (Table 1), a variable known to influence cross-modal temporal perception. The only way to investigate whether a difference between these groups is due to their different history of visual deprivation or their age is the comparison to an age matched healthy control group. The finding that both group comparisons revealed opposing effects, provides strong evidence for the role of vision following birth and against a role of unspecific factors such as the location of testing (the equipment was identical) for the reported group differences. We now describe this reasoning in further detail (Results paragraph one).

Our data indicate that DC-individuals show an impaired auditory and tactile resolution compared to their controls whereas these effects did not reach significance for the comparisons between CC-individuals and their normally sighted controls. Both groups differed from their controls with respect to bimodal and visual resolution (Results paragraph two).

We analyzed the correlation between resolution and bias. Participants with a lower temporal resolution relied more on temporal biases (Results paragraph three). This is reminiscent of the Bayesian framework in which the influence of the prior increases with decreasing reliability of the likelihood and makes intuitive sense: the less reliable the information extracted from the stimulus is, the more the brain relies on the readily available bias. This mechanism provides a potential explanation for the larger biases in DC-individuals compared to their controls (Discussion paragraph six). However, it cannot explain the main effect, the reversed direction of the bias in CC-individuals.

4) The MDC group didnt show the typical visual bias effect, this is a risk when using small sample sizes, and thus requires some consideration. Was this group not a representative control group? Again, based on Figure 1C it indeed appears that their performance at the bimanual condition was low. I appreciate that the interpretation of the DC group stands alone, by showing a bias effect in the vision direction (and opposite from the CC group). But still, this should be considered. Please don't address this question by showing no significant differences between the MDC and MCC groups, unless there are clear evidence to support this as lack of a difference (e.g. bayesian statistics)

The MDC-group did not show a significant bias in the proportion of “visual first” responses, yet, this does not allow us to conclude that they did not have a bias. Indeed, a closer look at the data suggests that the available power resulting from small sample sizes might play a role here. The apparent group difference is driven by one respectively two participants, which in a larger sample would not influence the statistical outcomes.

The sample sizes are too small to calculate reliable Bayes factors and collecting more participants for the control groups renders the comparison to the cataract-reversal groups problematic as the signal-to-noise ratio becomes skewed with unequal group sizes. We now show the distribution of biases across participants of each group using box-and-whisker plots (Figure 1D,E). This allows readers to convince themselves visually that both control samples show similar biases.

Finally, as stated in the comment, the key point for our study is the relation between the bias in the cataract-reversal groups compared to their matched controls.

Reviewer #2:Badde and colleagues presented a cross-modal temporal task (which stimulus was presented first) to controls, individuals born with dense, bilateral cataracts whose sight was restored after birth (CC), and those who developed cataracts after childbirth (DC). They found that those born with cataracts that were subsequently removed (the CC group) were more likely to respond visual 1st (vs. auditory or tactile) compared to the DC and matched control groups. The results are intriguing. However, there are major concerns with regards to the interpretation that diminish enthusiasm for these findings.The authors claim that "cross-modal perceptual biases are not innate but rather acquired during a sensitive period." My primary question is whether the observed effects are truly "cross-modal perceptual biases". An alternative hypothesis is based on the "law of prior entry", in which "the object of attention comes to consciousness more quickly than the objects which we are not attending to" (Titchener, 1908). The authors are aware of this, citing articles that show biases in temporal order when attending unequally across modalities (Titchener, 1908; Shore, Spence and Klein, 2001; Spence, Shore and Klein, 2001; Vibell et al., 2007). They then argue that (a) "attention has been ruled out as a cause of persistent biases in cross-modal spatiotemporal processing" and (b) "it is not obvious why CC- and DC-individuals would entertain opposing attentional foci given that both groups experienced transient severe visual impairment and suffer from remaining visual acuity impairments." For (a), I am not sure how their citation has ruled out attention as a cause of these biases from their Abstract: "we showed that temporal-order perception may be considered a psychological bias that attention can modulate but not fully eradicate". For (b), the argument would be that those w/o high-quality visual information at birth (CC) would need to attend more to the visual stimulus due to deficits/differences in their visual systems at birth, whereas those in DC group wouldn't need to, given that they were born with intact visual input.

We now discuss the possibility that congenital transient loss of pattern vision could lead to attentional biases that differ those of from normally sighted individuals and individuals with a later period of transient visual deprivation (Discussion paragraph five). In our (revised) view, the two accounts of the effect are not necessarily mutually exclusive: the pattern of sensory experience available at birth might calibrate modality-specific attentional biases as well as the underlying neural circuits recruited for temporal order perception.

To conclusively determine the role of attention for the observed reversal of cross-modal temporal biases, extensive new studies at the LV Prasad Eye Institute would be necessary. In order to collect data from a reasonable number of CC-individuals we would need 1-2 years under typical conditions. Participants for our studies are extremely difficult to recruit because we use strict selection criteria to maximize the likelihood that the tested CC-individuals were indeed born without any pattern vision. Currently under the condition of the pandemic it is unknown when we will be able to return to a typical testing schedule. Most of our participants come from rural areas and travel several hours in a bus to Hyderabad.

However, we think that our results are interesting irrespective of the role of attention for the effect. We can say for sure that whichever mechanism accounts for the reversal of cross-modal biases in CC-individuals, the perceived relative timing across modalities depends on early sensory (visual) experience during a sensitive period rather than, for instance, on visual experience later in life or on the current visual abilities. Thus, in the revised manuscript we relate our results to previously published studies, which seem to rather point against a shift of modality-specific attention towards vision in CC-individuals, but in the end due to the scarcity of studies directly targeting attention leave the possibility for such a mechanism open.

This is not to say that the author's hypothesis or this alternative hypothesis aren't interesting, but more data would be needed to tease these apart. This leads to my second issue, the paper is lacking in mechanism. The Introduction (especially for the lay reader) provides minimal information about why this bias occurs, and why this matters. The paper could be improved substantially with a more mechanistic rationale, especially in the Introduction.

We have rewritten large parts of the manuscript. The revised Introduction provides additional information about cross-modal temporal perception and describes mechanistic accounts of cross-modal temporal biases. The revised Discussion is more explicit about the mechanism which we consider being responsible for the reversed cross-modal bias in CC-individuals, and we now relate this “compensatory mechanism” to additional factors such as attention and response biases.

Finally, the authors note that the visual first bias in the CC group did not significantly correlate with duration of visual deprivation in CC group. Is this meaningful given the small number of subjects tested? It seems like there isn't enough evidence, as presented, to make this claim.

We have removed the correlational analysis.

Reviewer #3:The authors investigate cross-modal temporal perceptual biases in participants born with congenital bilateral cataract and treated on different ages. They find that in this population, the biases are opposite to healthy and developmental cataract populations. They conclude such biases develop in early sensitive period. The paper is intriguing and the results are important, however, it is to my view that the interpretation of the results and the main conclusion might be problematic.1) The authors main point is trying to rule out the possibility of innate cross-modal perceptual biases. The option of those biases being innate might still be possible, but biases might be reversed due to lack of visual input at early age while the brain is still flexible enough. This possibility should be mentioned. So, throughout the paper, I believe the claim of lack of innateness should be transform to that of sensitive period and innateness should only be discussed as a possibility in the discussion. If there are any developmental studies showing the emergence of the biases during the first few months after birth, they should be mentioned since they can support the authors' claim. The word "emerge" at the heading should also be change to something more accurate.

A stated in this comment, theoretically innate biases could be reversed during a sensitive period. Since we agree with Lewkowicz (2011, Infancy) that research on developmental principles should not be narrowed down to the simplistic dichotomy of nature – nurture, but rather investigate how experience dynamically contributes to brain development, we more precisely argue along these lines throughout the revised manuscript.

We now report findings which suggest that very young infants already show biases in the perception of cross-modal synchrony (Discussion paragraph three) and discuss parallels between the typical development of multisensory temporal perception and multisensory temporal perception in individuals with reversed congenital cataracts (paragraph nine).

We have changed the title of our manuscript.

2) In the Introduction: "Moreover, it is not obvious why CC- and DC-individuals would entertain opposing attentional foci given that both groups experienced transient severe visual impairment and suffer from remaining visual acuity impairments." The authors cannot use this claim to rule out attentional foci since it is also not obvious why CC and DC individuals will have an opposing cross-modal effect as well (this study demonstrates this exact result). Attention could be opposite for the same unknown reason. I can think of one reason why CC and DC groups will entertain opposing attentional foci. Developmental individuals were used to having high resolution input, now resolution decreased to medium so they are less attentive to vision. CCs were used to having low resolution input, now resolution increased to medium, so they are more attentive. This could be one possible reason why both groups will have opposing results in visual tests.

We now discuss the possibility that congenital loss of pattern vision leads to increased attention towards that modality (Discussion paragraph four).

3) DC group differed from the control group in the visuo-tactile experiment (with a trend in the visuo-auditory). Doesn't it mean that the sensitive period for the cross-modal perceptual biases is very long? Since those biases were altered (increased, not reversed) at a relatively late age? At least, this difference should be acknowledged in the Discussion acknowledging that there is no theory justifying this effect.

An increased bias towards perceiving vision as delayed in DC-individuals is incompatible with the idea of an extended sensitive period: if an extended sensitive period existed, the group difference between the DC- and the MDC-group should go in the same direction as that between the CC- and the MCC-group (since the DC- and CC-group share similarly altered experiences). Therefore, we hypothesize that the opposite cross-modal bias in the DC- compared to the CC-group fits well with the idea of a short sensitive period.

Unlike the normally sighted control group, DC-individuals might experience residual delays of visual processing for which the brain never learned to compensate because the sensitive period had closed early. An additional explanation for their increased cross-modal bias in the typical direction might be that DC-individuals rely stronger on typical biases because they experience greater temporal uncertainty. This idea is compatible with a common finding in individuals with restored hearing: Late deaf cochlear implant recipients show enhanced use of multisensory cues (Rouger et al., 2007, PNAS) while congenitally deaf cochlear implant recipients show no or impaired use of multisensory cues if the implants were received late (Schorr et al., 2005, PNAS). We now discuss this account in the manuscript (Discussion paragraph six).

4) Overall, following my previous comments, I think one can only conclude that cross-modal biases are sensitive to visual experience. If a stronger claim is needed, it should take into account both CC and DC groups' results.

We agree and now interpret the results as indicator for a sensitive period.